# Recurrent neural network dynamical systems for biological vision

**Wayne W.M. Soo**
Department of Engineering
University of Cambridge
wmws2@cam.ac.uk

**Aldo Battista**
Center for Neural Science
New York University
aldo.battista@nyu.edu

**Puria Radmard**
Department of Engineering
University of Cambridge
pr450@cam.ac.uk

**Xiao-Jing Wang**
Center for Neural Science
New York University
xjwang@nyu.edu

## Abstract

In neuroscience, recurrent neural networks (RNNs) are modeled as continuous-time dynamical systems to more accurately reflect the dynamics inherent in biological circuits. However, convolutional neural networks (CNNs) remain the preferred architecture in vision neuroscience due to their ability to efficiently process visual information, which comes at the cost of the biological realism provided by RNNs. To address this, we introduce a hybrid architecture that integrates the continuous-time recurrent dynamics of RNNs with the spatial processing capabilities of CNNs. Our models preserve the dynamical characteristics typical of RNNs while having comparable performance with their conventional CNN counterparts on benchmarks like ImageNet. Compared to conventional CNNs, our models demonstrate increased robustness to noise due to noise-suppressing mechanisms inherent in recurrent dynamical systems. Analyzing our architecture as a dynamical system is computationally expensive, so we develop a toolkit consisting of iterative methods specifically tailored for convolutional structures. We also train multi-area RNNs using our architecture as the front-end to perform complex cognitive tasks previously impossible to learn or achievable only with oversimplified stimulus representations. In monkey neural recordings, our models capture time-dependent variations in neural activity in higher-order visual areas. Together, these contributions represent a comprehensive foundation to unify the advances of CNNs and dynamical RNNs in vision neuroscience.

## 1 Introduction

Dynamical systems have long been a cornerstone in the field of neuroscience, tracing their origins back to the modeling of single neurons [1, 2]. Early biophysical models provided critical insights into the dynamic behavior of neurons. Building on this foundation, neuroscientists progressed to constructing networks of dynamical neurons [3–12]. The idea was to understand not just how individual neurons operate but how networks of neurons interact to produce complex behaviors [5–7, 12] and cognitive functions [8–10]. The rise of deep learning and artificial recurrent neural networks (RNNs) marked a significant turning point in this endeavor. Neuroscience quickly adopted RNNs due to their ability to model time-dependent processes [13–29], much like how biological circuits process information over time. In these models, each artificial neuron can behave according to some biophysical dynamics, whether through rate coding [13, 14, 18, 30] or spiking mechanisms [19, 20]. This alignment with

38th Conference on Neural Information Processing Systems (NeurIPS 2024).

biological plausibility made RNNs a powerful tool in neuroscience. Today, advances in computational power and learning algorithms have enabled these networks to be trained on diverse and complex tasks, ranging from motor control [14, 24] to cognitive functions [15, 16, 18, 27, 29–31]. Modern RNNs are capable of learning intricate patterns in data, making them invaluable for modeling a wide array of neural processes. We provide details of key references in Appendix A [1–5, 7, 13, 30, 32]. RNNs are not the only models used in neuroscience. Convolutional neural networks (CNNs) have also been widely used, particularly in the domain of biological vision [33–56]. CNNs excel at processing spatial hierarchies in images, making them ideal for object recognition and scene understanding tasks. While there are recurrent CNNs that integrate temporal dynamics into the spatial processing capabilities of CNNs [36, 38, 41–44, 49, 51, 57], the specific advances of dynamical RNNs in neuroscience have largely remained inapplicable to vision models. To bridge this gap, we propose a hybrid architecture that integrates the continuous-time recurrent dynamics of RNNs with the image processing capabilities of CNNs, which we refer to as CordsNet (**Co**nvolutional **R**NN **d**ynamical **s**ystem). Briefly, our contributions and results are:

- **Dynamical expressivity analysis.** We rigorously compare CordsNet with other recurrent dynamical architectures by training them all on multiple cognitive tasks in neuroscience. We find that CordsNet can achieve the same range of dynamical regimes as other architectures.
- **New training algorithm.** Training a continuous-time model is computationally expensive. We propose a computationally cheaper algorithm to efficiently initialize CordsNets and successfully train them on standard image classification benchmarks like ImageNet.
- **Autonomous and robust inference.** We show that our trained models can perform inference autonomously and exhibit robustness to noise compared to static and discrete-time CNNs, which are actually just properties inherent to dynamical systems.
- **Analytical toolkit.** There are many ways to analyze CordsNets, but they are memory intensive. We thus develop a toolkit consisting of iterative and dimensionality-reduction methods needed to analyze the model within reasonable computational limits and show some examples.
- **Image-computable models.** We demonstrate the effectiveness of CordsNet as the front-end of an image-computable multi-area model by training such models to perform tasks using the actual stimuli seen by subjects in experiments rather than abstract inputs.
- **Prediction of neural activity.** We find that CordsNets (trained on ImageNet) are able to predict temporal signatures in neural activity of higher-order visual areas (V4 and IT).

Together, our contributions represent a comprehensive and directed effort to bring decades of advancements in dynamical systems to vision neuroscience.

## 2 Model architecture

We first briefly introduce CNNs and dynamical RNNs in neuroscience. A typical convolutional layer in a CNN consists of a 2D-convolution, normalization (commonly batch normalization [58]) and a non-linearity (such as ReLU [59]), although the order of these operations can change [60]. For two consecutive convolutional layers $\mathbf{x}_{l-1}$ and $\mathbf{x}_l$, this can be written as:

$$\mathbf{x}_l = (\sigma \circ \text{Norm} \circ \text{Conv})(\mathbf{x}_{l-1}) \tag{1}$$

where $\sigma$ is the non-linear activation function. For a single image, the output of a convolutional layer is characterized by three dimensions (channels, height, and width), which can be interpreted as the latent state of the CNN (Figure 1A, left). On the other hand, a continuous-time RNN dynamical system typically found in neuroscience literature [28] is described by:

$$\mathbf{T}\frac{d\mathbf{r}}{dt} = -\mathbf{r} + \sigma \left(\mathbf{W}_{\text{rec}}\mathbf{r} + \mathbf{b} + \mathbf{W}_{\text{inp}}\mathbf{h}_{\text{inp}}\right) \tag{2}$$

where $\mathbf{T}$ represents a diagonal matrix of neuron time constants, $\mathbf{r}$ represents the latent state of the RNN, $\mathbf{W}_{\mathbf{rec}}$ is the recurrent weight matrix, $\mathbf{b}$ represents the vector of neuron biases, and $-\mathbf{r}$ is the leaky term which mimics the dynamics of biological neurons. $\mathbf{h}_{\text{inp}}$ is the external input to the network that first undergoes a linear transformation through $\mathbf{W}_{\text{inp}}$ (Figure 1A, middle). Just like in CNNs, $\sigma$ is the non-linear activation function. Note that there are no artificial normalizing operations as they are generally incompatible with continuous-time dynamical systems. The latent state of the network can be simulated forward in time by Euler's method with a suitable choice of time discretization.

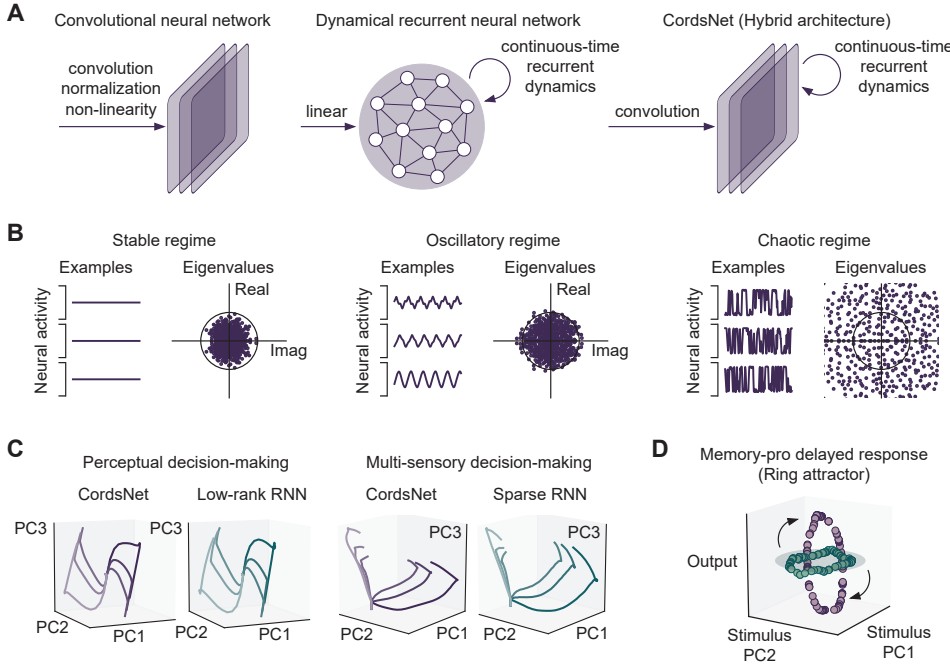

Figure 1: **A.** Overview of the proposed architecture (right) and its relation to CNNs (left) and RNNs (middle). **B.** Various dynamical regimes exhibited by randomly-initialized CordsNets. **C.** Aligned neural trajectories between a CordsNet and a low-rank RNN trained on a perceptual decision-making task (left), as well as between a CordsNet and a sparse RNN trained on a multi-sensory decision-making task (right). **D.** Example of a CordsNet trained to perform a memory-pro delayed response task by storing a circular variable in memory using a ring attractor.

Unlike regular RNNs, dynamical RNNs typically evolve (and are therefore backpropagated) over hundreds of time steps [28].

In CordsNet, which is our proposed hybrid architecture, every convolutional layer is replaced by a dynamical RNN (Figure 1A, right). The activations $\mathbf{x}_l$ now represent firing rates $\mathbf{r}_l$. In order to preserve the convolutional structure, the input and recurrent weight matrices ($\mathbf{W}_{\text{inp}}$ and $\mathbf{W}_{\text{rec}}$) are now convolutions ($\text{Conv}_{\text{inp}}$ and $\text{Conv}_{\text{rec}}$), while $\mathbf{h}_{\text{inp}}$ and $\mathbf{b}$ are reshaped to have convolutional structures. This can be summarized as:

$$\mathbf{T}\frac{d\mathbf{r}_l}{dt} = -\mathbf{r}_l + \sigma \left( \text{Conv}_{\text{rec}}(\mathbf{r}_l) + \mathbf{b} + \text{Conv}_{\text{inp}}(\mathbf{h}_{\text{inp}}) \right) \tag{3}$$

In the rest of this work, we will show that this proposed extension is highly non-trivial and brings the best (and worst) features of dynamical RNNs into CNNs.

## 2.1 Dynamical characteristics

Since convolutions are linear operations, they can be expressed as a 2-D weight matrix operating on a flattened 1-D vector, just like in equation (2), which we describe in detail in Appendix B.1. However, the convolutional recurrent weight structure of CordsNet spans only a small subspace of all possible recurrent weight matrices, so there is a need to verify if such a restriction would limit the range of dynamical properties that our networks can express. We know that (sufficiently large) Gaussian-initialized fully-connected networks using tanh activations exhibit different dynamical regimes based on initialization variance [4, 61]. We successfully replicate this in CordsNets to express stable, oscillatory and chaotic behaviors (Figure 1B) under the same conditions. Further analysis of dimensionality and autocorrelations can be found in Appendix B.2.

Multiple variants of dynamical RNNs have been trained on cognitive tasks in neuroscience [15, 28], and thus we want to know if CordsNets will produce similar solutions when trained on such tasks despite the restrictions from the convolutional weight structure. For this investigation, we choose

the same set of five tasks that was previously adopted for analyzing low-rank dynamical RNNs [62], consisting of a perceptual decision-making task [63], a parametric working memory task [64], a multi-sensory decision-making task [65], a contextual decision-making task [13] and a delayed match-to-sample task [66]. We independently train CordsNets, fully-connected RNNs, low-rank RNNs [62] and sparsely-connected RNNs [15] on these tasks across different activation functions, learning rates, network sizes and initializations. We find that trained networks of all architectures produce similar neural trajectories when aligned using canonical correlation analysis (Figure 1C), suggesting that CordsNets are able to utilize the same dynamical motifs as other established architectures in neuroscience to perform cognitive tasks. We also provide a more rigorous and quantitative analysis using recently proposed metrics on representational similarity [67] in Appendix B.3 which further supports this conclusion. In fact, from these metrics, we find that our networks produce more similar solutions to fully-connected RNNs than low-rank or sparsely-connected RNNs.

RNNs can perform tasks requiring long-term dependencies by storing information in memory. A dynamical RNN typically achieves this using attractors in neural activity space [68, 69]. We show that our networks can represent well-known classes of attractors such as ring (Figure 1D), line attractors, and discrete fixed-points (see Appendix B.4). Taken together, our results suggest that the convolutional recurrent weight structure of CordsNet does not constrain its expressiveness as a recurrent dynamical system.

## 3    Training and results

We now focus on the functionality of CordsNets as an image recognition model. We train networks of four different sizes, named CordsNet-RX, where $X \in [2, 4, 6, 8]$ represents the number of recurrent layers. Exact model specifications can be found in Appendix C.1, where we also review important design choices. We train our models on MNIST [70], Fashion-MNIST [71], CIFAR-10 [72], CIFAR-100 and ImageNet [73], each with dataset-specific augmentations [74] as detailed in Appendix C.2. The neuron time constant is set to 10 ms (constant for all neurons), and the network is simulated with 2 ms time steps. The loss function that we aim to minimize is computed as:

$$
\begin{aligned}
\texttt{loss} = &\texttt{logspace(-3,0,steps=30) * CEloss(output[170:200],labels)} \\
&+ \texttt{1e-3 * MSEloss(activity[290:300],spontaneous)}
\end{aligned}
\tag{4}
$$

We first simulate the networks for 200 ms (time steps 0 to 100) without any input so that they converge to some steady state spontaneous activity (`spontaneous`). A batch of images is then presented for 200 ms (time steps 100 to 200). During this time, the cross-entropy loss is computed for the last 60 ms (time steps 170 to 200) and combined using a log-weighted sum. Finally, the networks are simulated for another 200 ms (time steps 200 to 300), and the mean-squared error between activity in the last 20 ms (time steps 290 to 300) and spontaneous activity is added to the loss. This additional term encourages the networks to return to spontaneous activity after stimulus presentation. This loss function has been carefully designed to produce a mono-stable solution so that the network may perform accurate inference indefinitely across time, which we will elaborate on in the next section. We also provide a detailed ablation study of every coefficient and every term in Appendix D. Simulating these networks for 300 steps across time is computationally expensive. For comparison, we compute the total multiply-accumulate operations (MACs) for 79 well-known CNN models found in `torchvision.models` library and compare them against their parameter counts (Figure 2C). Our largest model, CordsNet-R8, has approximately the same number of parameters as ResNet-18, the smallest model of the ResNet series. In contrast, our smallest model, CordsNet-R2, requires more MACs to simulate than ViT-H-14, the largest vision transformer in the model library currently.

While training the models by computing and minimizing the loss function in equation (4) is inevitable, we can reduce the number of training iterations needed by carefully initializing our models. We do this in three computationally cheaper steps (Figure 2B). We first train a feedforward CNN model without the recurrent component and replace it with a one-time convolutional layer. We also include batch normalization here to improve training efficiency. In the second step, we fold the batch normalization [75] into the convolution operation and replace the one-time convolutional layers with dynamical linear RNNs. Batch normalization folding is done in the following way:

$$
\text{BN}(\mathbf{W}_{\text{conv}}\mathbf{r} + \mathbf{b}) = \underbrace{\frac{\gamma_{\text{BN}}}{\sqrt{\sigma_{\text{BN}}^2 + \epsilon}} \mathbf{W}_{\text{conv}}}_{\mathbf{W}_{\text{conv}}^{\text{fold}}} \mathbf{r} + \underbrace{\frac{\gamma_{\text{BN}}}{\sqrt{\sigma_{\text{BN}}^2 + \epsilon}} (\mathbf{b} - \mu_{\text{BN}}) + \beta_{\text{BN}}}_{\mathbf{b}^{\text{fold}}}
\tag{5}
$$

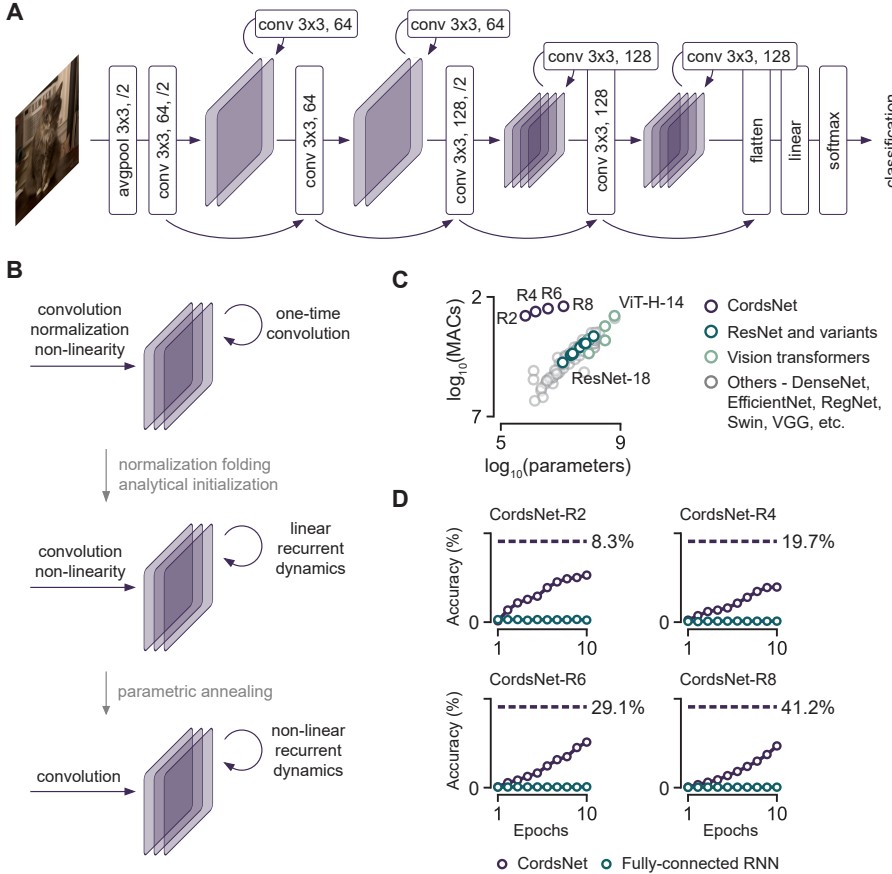

Figure 2: **A.** Architecture of CordsNet-R4. **B.** Proposed initialization method. A feedforward CNN is first trained (top). The parameters are then used to initialize and train linear RNNs (middle). Nonlinearity is then introduced by annealing (bottom). **C.** Multiply-accumulate operations (MACs) of 79 CNN models (green/grey) and CordsNet (purple) plotted against parameter counts. **D.** Validation accuracy of our models trained on ImageNet using the aforementioned three steps (dashed lines), compared to training CordsNet (purple circles) and fully-connected RNNs (green circles) directly.

A dynamical linear RNN is described by equation (2), except that $\sigma$ is simply the identity function. Such a network can be analytically solved for every time step $t$:

$$\mathbf{r}_t = e^{(\mathbf{W}_{\text{conv}}-\mathbf{I})t}\mathbf{r}_0 + \int_0^t e^{(\mathbf{W}_{\text{conv}}-\mathbf{I})(t-\tau)}(\mathbf{b} + \mathbf{W}_{\text{inp}}\mathbf{h}_{\text{inp}})d\tau \qquad (6)$$

We derive its steady-state solution, which can be approximated and compared to a convolutional layer:

$$\underbrace{\mathbf{r}_\infty}_{\text{conv output}} = (\mathbf{I} - \mathbf{W}_{\text{conv}})^{-1}(\mathbf{b} + \mathbf{W}_{\text{inp}}\mathbf{h}_{\text{inp}}) \approx (\mathbf{I} + \mathbf{W}_{\text{conv}} + \mathbf{W}_{\text{conv}}^2 + \dots)(\mathbf{b} + \underbrace{\mathbf{W}_{\text{inp}}\mathbf{h}_{\text{inp}}}_{\text{conv input}}) \quad (7)$$

The parameters of the linear RNN are $\mathbf{W}_{\text{conv}}$ and $\mathbf{b}$, which we optimize by minimizing the mean-squared error between its steady state $\mathbf{r}_\infty$ and the output of the convolutional layer, After optimization, we train the full linear model (end-to-end) using a reduced cost function:

```
loss_reduced = logspace(-3,0,steps=30) * CEloss(output[70:100],labels)   (8)
```

We present the image for 200 ms (time steps 0 to 100) and compute the weighted classification loss for the last 60 ms (time steps 70 to 100). We do not need to compute the spontaneous penalty term because linear RNNs are almost guaranteed to be mono-stable during training. Finally, in the third step, we replace the identity activation function with a parametric ReLU non-linearity [76]

Table 1: Test accuracies of CordsNets obtained using our initialization method and after fine-tuning, compared to their equivalent feedforward CNN counterparts. For controls, we trained CordsNets directly for the same time the initialization method (C) took. Fully-connected RNNs were also trained with matched parameter counts (R).

| Model | Dataset | Control (R) | Control (C) | Initialization | Fine-tuned | CNN |
|-------|---------|-------------|-------------|----------------|------------|-----|
| R2 | MNIST | 97.45 | 98.07 | 97.85 | 98.48 | 98.56 |
| | F-MNIST | 74.17 | 77.75 | 85.60 | 88.12 | 88.44 |
| | CIFAR-10 | 45.68 | 44.44 | 62.83 | 71.83 | 73.89 |
| | CIFAR-100 | 16.19 | 10.00 | 24.65 | 39.56 | 40.42 |
| | ImageNet | 0.22 | 5.55 | 8.27 | 14.23 | 15.57 |
| R4 | MNIST | 97.81 | 99.26 | 98.82 | 99.24 | 99.59 |
| | F-MNIST | 84.39 | 87.63 | 91.64 | 92.68 | 93.65 |
| | CIFAR-10 | 48.79 | 60.64 | 81.38 | 88.76 | 90.29 |
| | CIFAR-100 | 17.78 | 16.93 | 44.28 | 60.24 | 63.95 |
| | ImageNet | 0.31 | 11.67 | 19.67 | 33.78 | 36.28 |
| R6 | MNIST | 98.16 | 99.32 | 99.26 | 99.38 | 99.76 |
| | F-MNIST | 86.62 | 90.82 | 93.36 | 94.62 | 95.32 |
| | CIFAR-10 | 52.68 | 76.82 | 88.57 | 91.32 | 93.87 |
| | CIFAR-100 | 22.88 | 42.56 | 56.98 | 71.32 | 75.70 |
| | ImageNet | 0.22 | 19.33 | 29.14 | 50.14 | 52.06 |
| R8 | MNIST | 87.51 | 99.40 | 99.37 | 99.36 | 99.74 |
| | F-MNIST | 87.51 | 91.82 | 94.26 | 95.88 | 96.13 |
| | CIFAR-10 | 53.74 | 83.15 | 91.57 | 94.56 | 95.99 |
| | CIFAR-100 | 25.11 | 51.34 | 66.89 | 77.32 | 78.44 |
| | ImageNet | 0.26 | 23.21 | 41.24 | 57.90 | 63.16 |

with parameter $a$, where $\text{PReLU}(x) = \max(0, x) + a \times \min(0, x)$. We train the networks starting from $a = 1$ and gradually reducing $a$ until $a = 0$ at the end of training. This provides us with an initialization of the original model for further fine-tuning. We additionally perform two control experiments, where we train both fully-connected RNNs (with matched parameter counts) and CordsNets directly without our initialization method. In both controls, the number of training epochs is determined by matching the time required for CordsNets to be trained directly with the time required for CordsNets to be initialized using our method. Additional details about this time benchmark can be found in Appendix C.3. Test accuracies of all experiments can be found in Table 1 (for ImageNet, the validation accuracy is shown instead). We draw three main conclusions:

- **Initialization vs Control (C).** Our initialization method consistently produces models with higher test accuracies compared to models that were directly trained on every dataset except for MNIST, highlighting the effectiveness of our initialization approach. Figure 2D compares the ImageNet validation accuracy of CordsNets trained directly (purple circles) across all epochs against the validation accuracy of the models initialized using our method (purple dashed lines). MNIST classification is simple and does not require extensive feature extraction; it may therefore benefit from a more straightforward training approach.

- **Control (R) vs Control (C).** When trained for the same number of epochs, CordsNets R6/R8 consistently outperforms their parameter-matched fully-connected RNN counterparts on all datasets by a significant margin. Fully-connected RNNs perform remarkably poorly on ImageNet due to their poor scaling with image size (Figure 2D, green circles). However, the results become more ambiguous when it comes to smaller models on simpler datasets. We attribute this to the flexibility of fully-connected models, which allow them to attain early and fast gains in performance, especially when learning simpler features in smaller datasets.

- **Fine-tuned vs CNN.** Our fine-tuned CordsNets are able to attain accuracies that are slightly lower than (but reasonably close to) their feedforward CNN counterparts, suggesting that we have successfully trained a continuous-time dynamical system with competitive image processing capabilities. Closing the performance gap with CNNs remains a goal for future work.

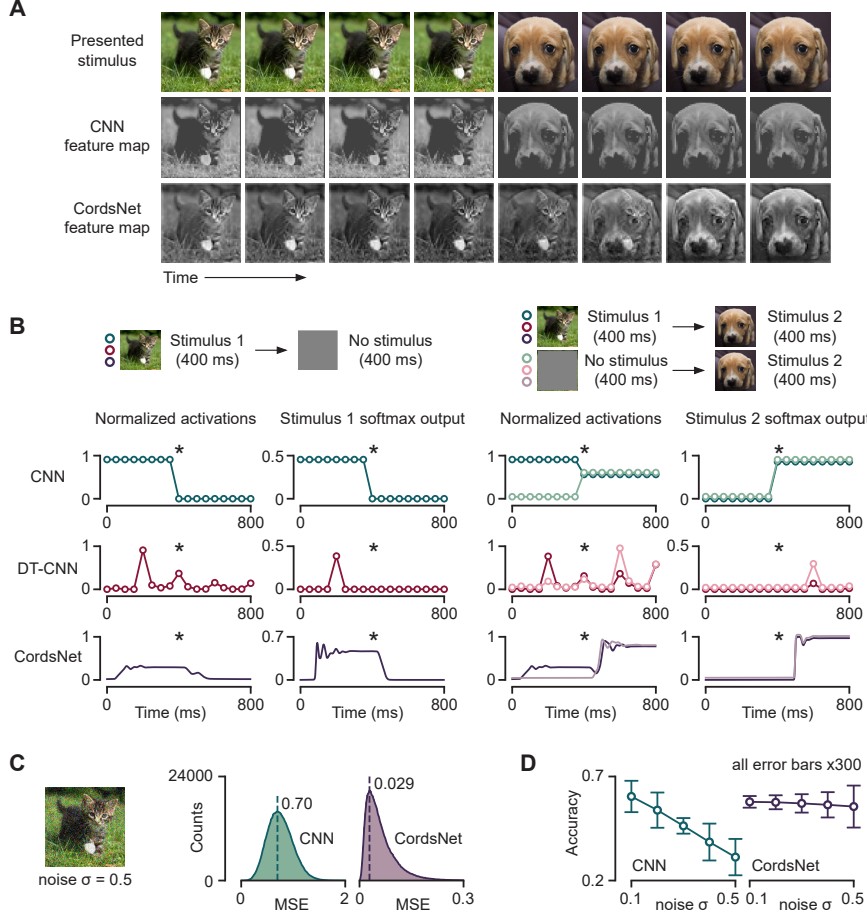

Figure 3: **A.** Evolution of a single interpretable feature map over time in the first layer of a feedforward CNN (middle) and CordsNet-R8 (bottom). **B.** Neural activity and softmax output of a feedforward CNN (green), discrete-time CNN (red), and CordsNet-R8 (purple) in response to various stimuli sequences. **C.** Mean-squared deviation from noiseless activations of the output layer across 50000 noisy images. **D.** ImageNet validation accuracies over 5 noise levels.

## 4    Model analysis

As a continuous-time model, the feature maps of CordsNets will change over time. We handpick a single interpretable feature map from CordsNet-R8 that depicts a gradual evolution over time when the input image is changed (Figure 3A). To emphasize the rich temporal dynamics that trained CordsNets express, we consider an additional comparison with CORNet-RT [38], a class of discrete-time recurrent CNN trained on ImageNet. In addition to architectural differences, the two models are trained differently. CordsNet has been trained to classify images over some time interval (and expected to perform accurately indefinitely across time), while CORNet-RT has been trained to classify for a single time step (and expected to be accurate for that particular time step).

We compare the temporal activities of the two models (as well as the feedforward CNN) in two scenarios. In the first scenario, an image is presented for 400 ms and then removed for the next 400 ms (Figure 3B, left). The feedforward CNN, having no concept of time, responds with the same activations across time and drops to zero when there is no input. The activity in CORNet-RT varies over time and spikes briefly at the time step, during which it is trained to accurately classify the input image. It is not able to make an accurate prediction at all other times. In contrast, the activity in CordsNet rises and stays at a fixed level for as long as the image is presented and returns to some baseline activity when the image is removed. In the second scenario, an image is presented for 400 ms, followed by a different image for 400 ms (Figure 3B, right). For its particular time step,

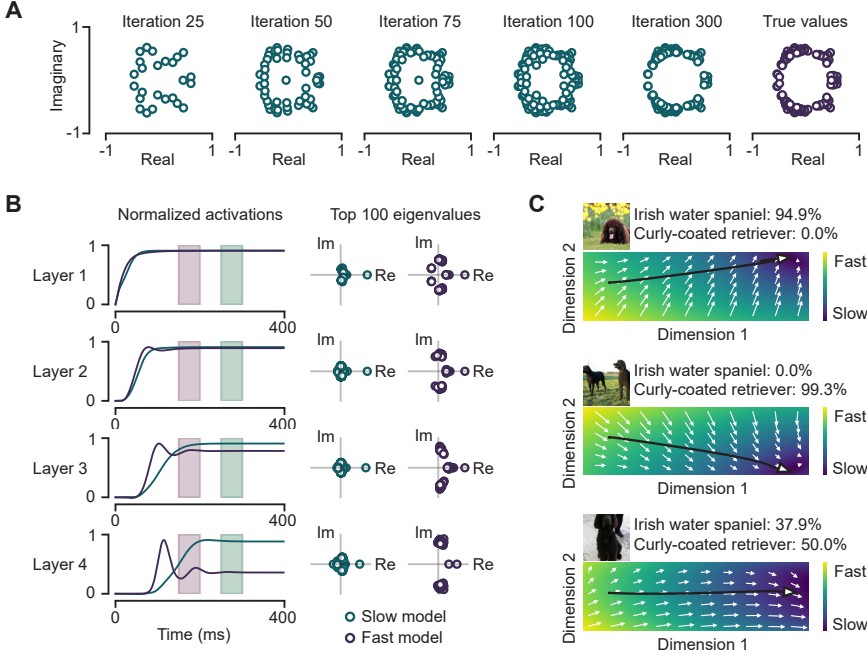

Figure 4: **A.** Arnoldi iteration [77] applied to a convolutional recurrent weight matrix. **B.** Model activations in CordsNet-R4 trained to classify images in time intervals $[140\,\text{ms}, 200\,\text{ms}]$ (purple) and $[240\,\text{ms}, 300\,\text{ms}]$ (green), along with their top 100 eigenvalues (right). **C.** Neural trajectories in the final layer of CordsNet-R8 projected onto two dimensions when presented with 3 different images.

CORNet-RT correctly classifies the first image but cannot correctly classify the second image, even though it can correctly classify the second image if it were presented first. CordsNet can correctly classify both images and once again maintains the correct classification throughout the duration of the stimulus. These results showcase the autonomous nature of CordsNet compared to discrete-time CNNs – it can perform inference indefinitely in time, self-reset to a baseline state, and react flexibly to stimuli changes, all while being governed by a single differential equation.

In the feedforward CNN, when temporally uncorrelated white noise is applied to the input image at each time step, model activations deviate from baseline noiseless values, which results in a decrease in classification accuracy (Figure 3C-D, green). When the same noisy images are instead presented to CordsNet, its deviations are attenuated by more than an order of magnitude compared to those of the feedforward CNN. It also robustly maintains its classification accuracy at high noise levels (Figure 3C-D, purple). This is a result of natural noise attenuation from continuous-time dynamics and filtering by the recurrent weights. We can rewrite the dynamics as an update equation to depict these effects:

$$\mathbf{r}_{t+1} = \left(1 - \frac{\Delta t}{T}\right)\mathbf{r}_t + \underbrace{\frac{\Delta t}{T}}_{\text{attenuation}} f(\overbrace{\mathbf{W}\mathbf{r}_t}^{\text{filtering (next step)}} + \mathbf{b} + \mathbf{W}_{\text{inp}}\mathbf{h}_{\text{inp}} + \text{noise}) \tag{9}$$

We next present an analysis of our trained models from a dynamical systems point of view. For a dynamical RNN, this typically involves performing some form of matrix decomposition on its recurrent weight matrix or applying dimensionality-reduction techniques on neural trajectories. For CordsNet, both of these approaches are particularly challenging due to the size of the networks. Memory limitations prevent the full recurrent weight matrix from being expanded from its kernel representation. One solution that we found is to compute the eigenvalues of the recurrent weight matrix directly from its kernel form using Arnoldi iteration [77] (Figure 4A), which we use to uncover an important dynamical motif present in our trained networks. We notice that the dynamical characteristics of our networks are different when we train them to correctly classify images at different times. When trained to classify early, the networks exhibit an oscillatory behavior, but not when trained to classify late (Figure 4B, left). We attribute this to the effect of transient overshooting,

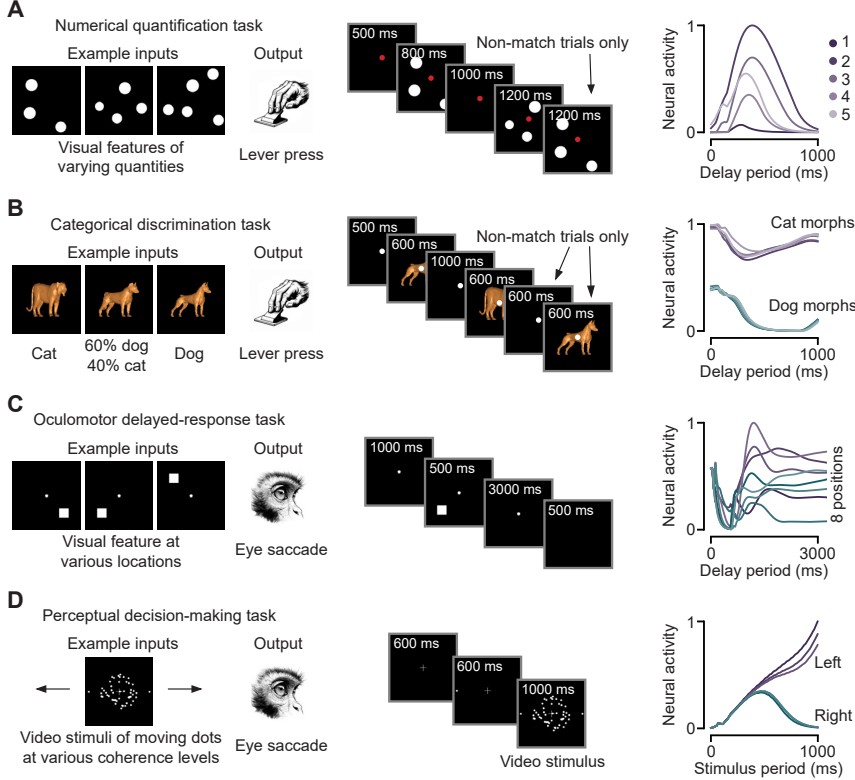

Figure 5: A multi-area model consisting of CordsNet-R8 connected to a fully-connected RNN trained on various cognitive tasks in neuroscience. **A.** Numerical quantification task where monkeys are required to remember the number of visual features on the screen. **B.** Discrimination task where monkeys must discern whether an image is predominantly cat or dog. **C.** Delayed-response task where monkeys have to saccade to the location of a previously shown stimulus. **D.** Evidence integration task where monkeys are required to tell if random dots are moving left or right.

where being in the oscillatory regime results in an early overshoot in activity, thereby speeding up the propagation of the signal. In the linear approximation, according to equation (6), this would manifest if the eigenvalues of $(\mathbf{W}_{\mathrm{conv}} - \mathbf{I})$ of the model trained for faster inference contain larger imaginary components. We confirm this to be the case (Figure 4B, right). It is also possible to perform dimensionality reduction within reasonable memory limits, but these methods require some adaptation for convolutional operations. Therefore, we release a toolkit consisting of the aforementioned functions that are specifically tailored for convolutional weights. These convenient tools unlock many possible approaches for analyzing CordsNets. For example, we identify a particular dimension in CordsNet-R8 which represents whether the model perceives an image as an Irish water spaniel or a curly-coated retriever (Figure 4C).

## 5   Applications

CordsNet is an ideal front-end of any RNN dynamical system that requires image-computability. To demonstrate this, we connect the final layer of CordsNet-R8 with a fully-connected RNN (with 512 neurons) and train only the fully-connected RNN on a set of cognitive tasks that explicitly requires visual information processing (Figure 5). We use the actual images the monkeys see as the input to our multi-area model. This is done either by obtaining the stimuli set from the authors of the experiments [78, 79] or generated according to the specifications described in the original experimental papers [80, 81]. In our trained models, we found interpretable neurons in the fully-connected RNN layer, such as neurons tuned to stimulus quantity (Figure 5A, right), cats or dogs (Figure 5B, right), spatial positions on the screen (Figure 5C, right) and motion direction (Figure 5D, right).

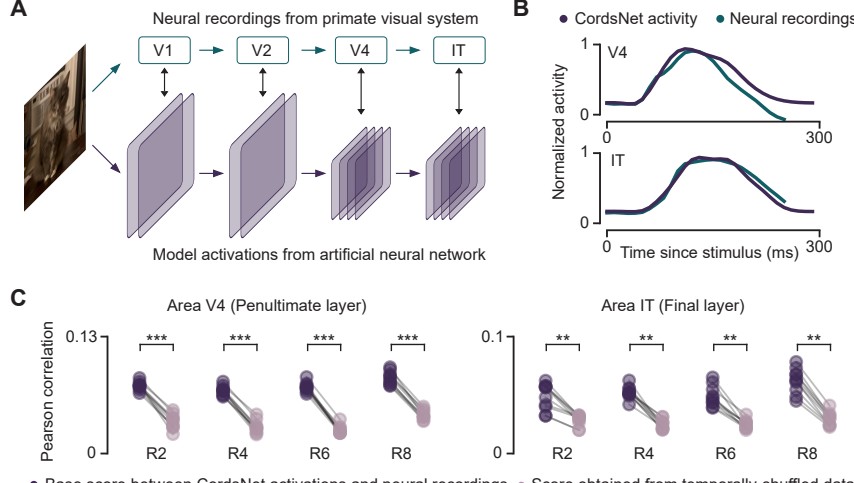

Figure 6: **A.** Framework of Brain-Score (Vision). **B.** Normalized CordsNet-R8 activity in the last two layers compared with experimentally-recorded [82] normalized activity in visual areas V4 and IT. **C.** Similarity metrics between CordsNet model activations and neural data (dark purple) and temporally-shuffled neural data (light purple). (paired t-test, ** $p < 10^{-4}$, *** $p < 10^{-7}$)

Brain-Score is a benchmarking framework designed to evaluate the performance of artificial neural networks in terms of their ability to model and predict neural and behavioral data from the brain [39]. In particular, model activations from CNN layers have been compared with time-averaged neural data from various parts of the primate visual system (Figure 6A). With our continuous-time model, we want to know if our models are able to model temporal signatures in brain activity. To do so, we omit the time-averaging step when computing Brain-Scores (see Appendix E for details). Using the same training-test splits, we additionally fit the same model activations with temporally-shuffled neural data. Due to the lagged response in neural data, we have to temporally shift our model activations such that the time in which activity rises in response to an input is aligned with neural data. There are occasions where no shifting is necessary, such as in the final two layers of CordsNet-R8 (Figure 6B) on V4 and IT neural activity [82]. All CordsNet models score higher on unshuffled data across V4 and IT (paired t-test, all p-values $< 10^{-4}$, Figure 6C), suggesting that the models are capturing temporal structures within the neural data. For the sake of transparency, we also state that we have performed the same test on a different dataset with V1 and V2 activities [83] but obtained inconclusive results.

## 6 Discussion and conclusion

We have presented CordsNet, a hybrid architecture combining the strengths of CNNs and dynamical RNNs to process visual information with continuous-time dynamics. CordsNet exhibits various dynamical behaviors, such as oscillations and chaos (Figure 1B). In our analysis of its image processing capabilities, we showcase its ability to classify images indefinitely across time (Figure 3B, right), as well as its robustness to noise (Figure 3D, right). We have also effectively used CordsNet as a means to build image-computable dynamical systems capable of performing cognitive tasks (Figure 5). Finally, CordsNet has been successful at modeling the temporal signatures of neural activity in higher-order visual areas like V4 and IT (Figure 6C).

**Limitations.** The main limitation of CordsNet lies in its substantial memory requirements (Figure 2C). Memory-efficient training algorithms have been explored [84]. Another limitation is the difficulty of analyzing CordsNet as a dynamical system, primarily due to how it is unfeasible to convert the recurrent kernel weights into its corresponding full-size recurrent weight matrix.

**Conclusion.** CordsNet bridges a crucial gap between dynamical systems and vision neuroscience. While most of cognitive neuroscience continues to build upon decades of RNN research, vision neuroscience remains dominated by CNN models due to the inability of RNNs to efficiently process visual information. CordsNet has resolved this limitation by being a dynamical RNN with image processing capabilities.

## Author contributions

W.S. designed the architecture. W.S, A.B. and P.R. trained and analyzed the networks. All authors designed the study, took part in discussions, interpreted the results, and wrote the paper.

## Acknowledgments and Disclosure of Funding

This work was supported by the NIH grant R01MH062349, Office of Naval Research grant N00014, James Simons Foundation grant NC-GB-CULM-00003138 and NYU High Performance Computing. A.B. was supported by the Swartz Foundation.

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
