# OpenReview forum: "Recurrent neural network dynamical systems for biological vision"
_NeurIPS.cc/2024/Conference — NeurIPS 2024 spotlight_

### Official Review · Reviewer_UEUJ · 2024-07-11

**Soundness:** 3
**Presentation:** 3
**Contribution:** 2
**Rating:** 6
**Confidence:** 2

**Summary:**

The paper proposes a hybrid architecture that integrates continuous-time recurrent neural networks (RNNs) with convolutional neural networks (CNNs), named CordsNet, to improve biological realism in vision models. The authors claim that CordsNet matches CNN performance on benchmarks like ImageNet while showing increased robustness to noise. They also present a toolkit for analyzing these models and demonstrate the model's ability to capture time-dependent neural activity.

**Strengths:**

1. The proposed model demonstrates connections to neural behavior and shows effectiveness in dealing with continuous-time modeling. It might inspire future biologically feasible network design.
2. The sharing of weights for convolutional and recurrent operations is novel since these are typically done sequentially in a backbone.
3.  The introduction of tools for analyzing convolutional structures in dynamical systems is potentially helpful for future research, but more details are needed.

**Weaknesses:**

1. The comparative result can be conducted more comprehensively. Some comparative results are not conducted on models with the same temporal processing power (Figure 3 only CNN shown).  Comparison to similar temporal models such as convolutional RNN  and vanilla RNN are lacking. These models can likely exhibit similar behaviors while requiring a less complex training scheme.
2. The choice of image datasets is questionable. The dataset is in discrete-time space, and continuous-time training might not provide any advantages over traditional CNN. The proposed model enables intra-batch information flow between images, which can provide certain robustness to discrete data. Still, the advantages should be more prominent when images in a batch are naturally sequential and have dense temporal information.

**Questions:**

What’s your model’s advantage over a simple sequential design of convolutional preprocessing followed by a dynamical RNN?

**Limitations:**

1. The use case of the proposed model is limited and the advantage over discrete-time models is not clear. I believe continuous-time vision tasks are needed to fully demonstrate the effectiveness of this method. Tasks such as video prediction, and video feature extraction might be good candidates. Small-scale experiments can be also conducted on augmented image datasets with more temporal information.
2. The proposed model is novel, but if it provides advantages over existing hybrid models remains questionable. It shares similar design intuitions to convolutional RNN, effectively mimicking the visual preprocessing of visual information followed by temporal processing in a later stage in the brain. Such designs are widely used in the form of CNN + vision Transformer in video processing (XMem, Cheng & Schwing, 2022) and video generation (Video Diffusion Model, Ho et al., 2022) domain where both spatial and temporal processing are conducted concurrently.  However, a comparison with such an existing design is not provided. The provided CNN baseline is relatively weak compared to the above-mentioned recent architecture.
3. Intuitively the spontaneous activity attraction is functionally similar to weight decay, whether this regularization term provides advantages remains questionable. Ablation studies can be conducted to provide more rationales behind your design choice since multiple design options are present in your model.

---

> ### Author Rebuttal · Authors · 2024-08-07
>
> We thank the reviewer for their time and insightful suggestions. For the sake of consistency across reviews, we would first like to clarify a few definitions used in our response:
> - We think that the reviewer is referring to "convolutional RNN" as a CNN connected to an RNN (typically LSTM in literature but could be any type of RNN). This is most likely the widely-adopted definition that the reviewer is using. However, in our response, we will refer to these models as CNN-RNN models.
> - In contrast, we will refer to convolutional RNNs as RNNs whose recurrence is represented by a convolution, just like in this work. This is also consistent with the terminology used by other reviewers.
>
> **Advantages over CNN-RNN**
>
> In the first point raised in the weaknesses section, as well as in the question section, the reviewer made a sharp observation that simply using CNN-RNN architectures would be sufficient to reproduce the results in Figure 3, and therefore questioned the advantage of our proposed model. We raise several orthogonal points below that we hope would help the reviewer see our work in a better light.
>
> - **Not all results can be reproduced by CNN-RNN.** We agree that the results in Figures 3 and (quite likely but we cannot confirm) Figure 5 can be obtained by the CNN-RNN architecture, but the results in Figures 4B and 6 remain exclusive to our continuous-time model. The phenomenon in Figure 4B is attributed to the temporal lag that arises due to every layer being continuous in time and require time to ramp up. Figure 6 compares neural trajectories across time in CNN layers with neural activity in the macaque ventral stream (their biological visual system), but the CNN in a CNN-RNN architecture is static in time.
>
> - **Applying results from CNN and RNN neuroscience.** CNNs and RNNs have both been proposed as models of the brain (or parts of it). By building a hybrid model, we may integrate results from both fields which may lead to unified results and potentially generate new hypotheses about how the brain works.
>
> - **Redefining expectations for this work.** If the end goal of this work is to find a model that can perform the feats that we show in the paper, then CNN-RNNs would probably have an edge over our model.  But our goal here is to build models of the brain and help improve our understanding of how continuous-time dynamics work in a competent model of vision. This means that we incorporate biological constraints in our models, even if they are detrimental to final performance. Just like how the brain operates in continuous-time, including its visual system, we build a model that also runs continuous in time.
>
> - **CNN-RNNs requiring less complex training.** We absolutely agree that the computational resources required to train a model with a static CNN is considerably less than the resources required to train CordsNet. We like to view this as a positive aspect of our work, rather than a negative one. We are knowingly building a model that is harder to train, with constraints that are likely to impede training and testing performance (short inference time, continuous-time and recurrent properties offering nothing to image classification performance), so that we can obtain a model that can be used to study the dynamics underlying visual processing in a continuous-time dynamical system. In fact, we consider the success of our training, using our proposed algorithm, to be a key result of this work.
>
> **Suitability of ImageNet and other use cases**
>
> The reviewer has raised concern that ImageNet, being a static dataset that does not change in time, may not be the most suitable task for showcasing how our model works. We agree with this viewpoint and are impressed by their pursuit for engineering optimality. Fundamentally, the theoretical and experimental sides of neuroscience work closely with each other. New experiments are conducted from proposed theories, and in turn new models are built from experimental results. Similarly, this work is designed based on influences from a variety of past works in neuroscience. There has been decades of results on dynamical models of the brain, which are still being worked on today. In recent years, CNNs have gained traction in vision neuroscience, and one of the main driving forces of this is the BrainScore platform [(link here)](https://www.brain-score.org/vision/), where CNNs are trained on ImageNet, and their neural activities are then compared to neural data. This also comes with years of results reported by various neuroscience groups. Our goal here is to bridge the gap between results from dynamical system models and results from CNN models for vision. To that end, training and evaluating our model on ImageNet would best facilitate this.
>
> **Spontaneous activity penalty**
>
> We thank the reviewer for the suggestion. We have performed ablation studies in the general response that would help clarify your doubts on the impact of this term.
>
> We would also like to specifically comment on the reviewer's intuition, which is largely in the correct direction. We cannot make strong claims for non-linear models, but in the linear case of our model, it would be the largest eigenvalue of the recurrent weight matrix that would determine whether the model is monostable or not. This is definitely related to the magnitude of the weights to some extent, just as the reviewer has suggested.

---

> > ### Comment · Reviewer_UEUJ · 2024-08-11
> >
> > Thanks for the clarification and the additional file.
> >
> > I'm not very familiar with the related works in the neuroscience community, but based on the explanation and the reviews from other reviewers, I believe this work has a moderate amount of contribution in its field and will adjust my score up by a point.
> >
> >  I have some personal questions from a deep-learning perspective, that might or might not related to this work:
> >
> > 1. The goal of this work is to close the gap between artificial neural networks to neural activities(?), is the backpropagation-based optimization inherently unsuitable for this goal?
> > 2. It looks like transformer variants (CVT) have great performance on BrainScore, is your method potentially extendable from recurrence dynamics to self-attention dynamics?

---

> > > ### Author Response · Authors · 2024-08-13
> > > **Thank you**
> > >
> > > We thank the reviewer for their revised score and we are very happy to see that the reviewer is curious about potential future directions of this work.
> > >
> > > *(For meta-reviewing purposes, we would like to state that everything below is not related to the rebuttal and is simply an out-of-scope discussion with the reviewer.)*
> > >
> > > Backpropagation is generally understood to be biologically not realistic, and the reviewer is right to doubt whether models trained with backprop are appropriate for the pursuit of uncovering the mysteries of the brain. Plasticity (the neuroscience term for learning by changing connection weights) is a big field in neuroscience, and our line of work would greatly benefit from implementing a biologically realistic learning mechanism. Right now, backprop remains superior not just for its simplicity but also due to its optimization in hardware by NVIDIA. As such, for now we can argue that we are just interested in the end goal (a trained model that can do some task), and how we get there (backprop) is not too important. See [Table S1](https://proceedings.neurips.cc/paper_files/paper/2023/file/65ccdfe02045fa0b823c5fa7ffd56b66-Supplemental-Conference.pdf) here for a list of influential RNN models in neuroscience, trained by backprop.
> > >
> > >
> > > The reviewer also aptly raised an interesting point that transformer architectures are also doing well on BrainScore. From a neuroscience point of view, visual information processing typically happens in the visual areas of the brain. Biological attention, maintaining fixation, and general interpretation of the visual signal is done brain-wide, and commonly studied in the prefrontal cortex (which includes the frontal eye fields) along with their top-down feedback loops with the visual pathway [(example)](https://www.nature.com/articles/s41586-021-03390-w). We speculate that it would be more appropriate for self-attention mechanisms in vision to be implemented as another brain area, and the interesting way forward would be to understand the interactions between the two models of the two different areas.

---

### Official Review · Reviewer_7SU8 · 2024-07-11

**Soundness:** 4
**Presentation:** 4
**Contribution:** 4
**Rating:** 9
**Confidence:** 4

**Summary:**

Biological neural networks are continuous-time dynamical systems. However, the previous models that are carefully designed to explain biological networks are discrete-time systems or even non-dynamical ones like convolutional neural networks (CNNs). However, the model that best explains biological vision and achieves high performance on downstream tasks is CNNs. This work fills the gap by creating novel networks that combine CNNs and continuous-time recurrent neural networks called CordsNet. CordsNet is evaluated on a standard ImageNet classification task and also across several standard cognitive neuroscience tasks.

**Strengths:**

1. Biological realism during inference: CordsNet might be one of the most biomimetic deep learning models, as its dynamics are more realistic than other state-of-the-art models like CORnet-RT and CORnet-S. It is motivated by neural dynamics models.

2. A broad range of cognitive tasks are studied: Beyond the ImageNet classification task, this work investigates the responses of CordsNet on several standard cognitive tasks that neuroscientists use to study humans and animals.

3. Benefit of recurrence dynamics over CNN: CordsNet demonstrates the advantage of classification with noisy images over CNN. This suggests the missing piece to make computer vision models more robust, like humans.

4. Great visualization: The visualization is very insightful and provides intuition on how CordsNet works.

**Weaknesses:**

1. Other methods like CORnet-RT and CORnet-S are not evaluated on the cognitive tasks that plug RNN. Therefore, we cannot see if CordsNet is better than others on this benchmark. I believe they can perform the same tasks but they may have different characteristics or different representations than CordsNet

2. CordsNet is trained to predict stimuli at a certain duration, but it cannot accurately predict other durations.

3. The task is supervised image classification where the models learn from rich human labels. Moreover, brains perform with minimal or no strong supervision from others. Instead, brains may learn in a self-supervised manner. It is possible to explore self-supervised learning tasks that are more like what brains do.

4. Embodied tasks are not included: To study the benefits and characteristics of neural dynamics, embodied tasks may be more interesting.

**Questions:**

1. Citation [36] is wrong. That one is CORnet-S, not CORnet-RT [1].

2. Why are brain scores low compared to the leader board on the same public datasets? Is the score a neural predictivity score defined in the Brain Score paper [2] or a different metric?

3. How are brain-scores of the proposed method compared to CORnet-S and CORnet-RT? How many total parameters of the CordsNet compared to those models?

Reference

[1] Kubilius, Jonas, Martin Schrimpf, Aran Nayebi, Daniel Bear, Daniel LK Yamins, and James J. DiCarlo. "Cornet: Modeling the neural mechanisms of core object recognition." BioRxiv (2018): 408385.

[2] Schrimpf, Martin, Jonas Kubilius, Ha Hong, Najib J. Majaj, Rishi Rajalingham, Elias B. Issa, Kohitij Kar et al. "Brain-score: Which artificial neural network for object recognition is most brain-like?." BioRxiv (2018): 407007.

**Limitations:**

A lot if memory is required during training.

---

> ### Author Rebuttal · Authors · 2024-08-07
>
> We are heartened by the extremely positive review and high regard the reviewer has for our work. We thank the reviewer for their time and encouragement.
>
> **Comparison of CORNet on tasks in Figure 5**
>
> We agree with the reviewer that CORNets could potentially arrive at different solutions compared to CordsNets. This would be in line with our efforts to bring analyses that are commonly found in dynamical systems (such as the cognitive tasks in Figure 5, where a dynamical RNN would have to be plugged) to the CNN vision neuroscience community. It is exactly our hope that our analysis of CordsNet as a dynamical system would spark such applications for CORNets and other models in vision neuroscience.
>
> **Performance across time of CordsNet**
>
> We note that since CordsNet arrives at a steady-state at time of inference, it can indefinitely maintain the accurate classification for as long as the image is still presented, even far beyond the trained duration. In addition, CordsNet is also able to reset back to baseline when an image is removed, and correctly classify another new image when it is presented at a later time. These results can be found in Figure 3.
>
> **Self-supervised learning and embodied tasks**
>
> We fully agree with the reviewer that the literature of CNNs in neuroscience have progressed far beyond straightforward supervised learning. As a starting point for incorporating dynamical system, we believe that our work will motivate future endeavors on these ideas (including for ourselves).
>
> **Citation error**
>
> We thank the reviewer for the clarification.
>
> **BrainScore clarification**
>
> In order to evaluate our models' ability to capture temporal signatures in neural data, we had to extend the BrainScore similarity metric to account for time.
>
> The original formulation performs a partial-least-squares fit between CNN activations, with shape [number of images, number of nodes] with neural data, with shape [number of images, number of neurons]. This is done with time-averaged neural activity.
>
> Here, we make the most minimal extension, so that we are now fitting CNN activations, with shape [timesteps $\times$ number of images, number of nodes] with neural data, with shape [timesteps $\times$ number of images, number of neurons], which means that the time-averaging step is omitted. Without the time averaging step, the BrainScore drops.

---

> > ### Author Response · Authors · 2024-08-13
> > **Thank you**
> >
> > As the discussion period is coming to an end, we would like to once again thank the reviewer for the positive endorsement of this work, and for taking part in the rebuttal with other reviewers.

---

### Official Review · Reviewer_7DkU · 2024-07-12

**Soundness:** 2
**Presentation:** 2
**Contribution:** 2
**Rating:** 5
**Confidence:** 2

**Summary:**

The paper proposes a recurrent convolutional neural network architecture and training algorithm that simulates the biological visual system of mammals. The technical novelty is mostly in the training algorithm which is meant to mimic biological systems, with a first stage of spontaneous activities followed by learning and then again spontaneous activities. Since this process is too computational intensive, the authors introduce approximations, such as initializing from a supervised trained network. The empirical analysis shows that the method is not too far from standard supervised learning while being robust to noise and while exhibiting a behavior that better matches biological systems. Also, the approach is used to predict brain activity data as an application.

Disclaimer: My background is machine learning rather than comp. neuroscience. I am not qualified to assess the impact that this work can have in that community and my assessment is mostly limited to the ML side of this contribution.

**Strengths:**

+ simple and intuitive architecture
+ generally well written and fluent paper
+ the loss definition (in its original formulation of eq. 2) is very interesting and novel
+ nice analysis of how this work relates and contributes to the field of comp. neuroscience
+ overall motivation and research topic

**Weaknesses:**

- important technical details are missing. For instance, in sec. 2 only few lines are used to give a high level description of the architecture. Too much material that should be in the main paper is placed in the appendix (e.g., comparison to supervised CNNs mentioned in the abstract) . Overall the paper is not self-contained and misses critical details.
- there is a big approximation between the original loss function and what actually is minimized in practice. If the goal is to simulate a biological system, it is unclear to me how the approximation made fit in the context of that goal.
- overall, claims are not entirely supported. For instance, it is not true that on ImageNet the performance is "comparable" to a standard CNN. A difference of 5% is big in that context. Some accomplishments of this work should be toned down a bit.
- missing references: There is prior work on recurrent CNNs that is not cited. For instance,
Recurrent Convolutional Neural Network for Object Recognition by Liang et al. CVPR 2015 uses the same architecture as far as I can tell.
- missing ablations: The authors well ablated the contribution of the recurrent and convolutional part of their model. However, their approach makes lots of design choices which are not very well justified empirically. For instance, what happens if fewer recurrent iterations are used? or what happens if only the top-most fully connected layer are made recurrent? what happens if some terms of the loss are removed? etc.

**Questions:**

I am curious whether the model oscillates when there are multiple interpretation of the input. In the simplest setting, the input could be linearly "mixed-up". I am also curious whether "easy" examples require less iterations to converge.

**Limitations:**

No concern.

---

> ### Author Rebuttal · Authors · 2024-08-07
>
> We thank the reviewer for the helpful suggestions. In response to these points, we have performed additional analyses and made several changes to our submission, as explained below (or referred to general response).
>
> **Important information in appendix**
>
> We thank the reviewer for raising this important point. We would like to direct the reviewer to our general response, which provides a detailed plan on how we plan to rebalance the results in the main text and the appendix.
>
> **Issue of unsupported claims**
>
> We agree with the reviewer that the performance of CordsNets fine-tuned on ImageNet is lower than their CNN counterparts with the same convolutional layers, and that this discrepancy may possibly be large enough to challenge our claim. We have made the necessary changes to the submission to account for this, as detailed below.
>
> - Line 163 updated to read, "We also find that our fine-tuned models perform only slightly poorer than their initial feedforward CNN counterparts, which nonetheless represents a significant improvement over existing continuous-time dynamical models in visual processing."
> - Line 387 updated to read, "Our fine-tuned CordsNets are able to attain accuracies that are only slightly poorer than their feedforward CNN counterparts."
>
> We feel that the term "comparable" is modestly subjective. We initially chose this word because of the current inability of continous-time dynamical systems to process visual information, as seen in the third column of Table S5. Dynamical RNNs are barely beating chance level in a classification task with 1000 classes, which is rightfully "incomparable" to CNNs. By being in a very broad performance range (50% to 80%), we feel that we may use the term "comparable" in this very specific context. However, we will still make the changes above for clarity.
>
> At the same time, we wish to also review and defend the other claims that we have made about this work, which we hope that the reviewer would agree with.
>
> - **Dynamical expressivity analysis.** We claimed that we have rigorously explored the possible dynamical regimes that our model can express, and also made comparisons to other model architectures. We provided instances of different dynamical regimes in Figure 1B. In addition to CordsNet, we also trained low-rank, dense and sparse RNNs (Section B.3, which is in the appendix in the initial submission) on five different cognitive tasks in neuroscience (Section B.4), for three different network sizes (Section B.5). We then compared the activity trajectories of all these trained models (results from Figures S2 and S3).
> - **New training algorithm.** We have provided full details of our training algorithm in Section 3 in the main text. In addition, we explain certain design choices and perform ablation studies for the loss function in the general response.
> - **Autonomous and robust inference.** We show in Figure 3B that our model is able to reset to baseline levels after stimulus presentation, and can then accurately classify a new input image. This supports our claim on autonomous inference. We also provide evidence that our model is robust to noise, due to the "evidence integration" mechanism that is present in dynamical systems. This is explained in lines 190-196, and equation (7).
> - **Analytical toolkit.** We claim to provide a toolkit consisting of Arnoldi and power iteration algorithms and partial SVD specifically for convolutional architectures (i.e. our model). We demonstrate the effectiveness of our implementation of Arnoldi iteration in Figure 4A, and applied partial SVD to a particular scenario in Figure 4C.
> - **Image-computable models.** We have trained CordsNet appended to a fully-connected RNN on four actual tasks in neuroscience literature (Figure 5, left), and provided evidence of our trained models (Figure 5, right). At the same time, we acknowledge that this analysis is brief, which is why we make no predictions or neuroscientific claims based off our results here. An in-depth investigation on this topic would be outside the scope of this work. The intention is to show the potential applications of CordsNet as a multi-area, image-computable model, which we have delivered.
> - **Prediction of neural activity.** We leveraged on a robust and popular benchmark platform known as BrainScore to compute similarity metrics between model activations in CordsNet to actual neural activity recorded in the visual system of the macaque monkey (Figure 6). We show statistical significance in our results.
>
> We hope that with this overview, the reviewer would agree with the stated accomplishments of this work.
>
> **Missing references**
>
> We thank the reviewer for the citation. and thoroughly comb through relevant literature in our final version.
>
> **Motivation for loss function and ablation studies**
>
> We thank the reviewer for this suggestion. We have done an extensive ablation study for our proposed loss function, as detailed in the general response.
>
> **Model behavior subjected to ambiguous input**
>
> The concept of multistable perception has been extensively studied in neuroscience, with many theories proposed based on dynamical systems. Coincidentally, we have shown preliminary results on this phenomenon in Figure 4C, where the activity just goes to some interpolated activity space between the two interpretations, which is currently still a steady-state response.

---

> > ### Comment · Reviewer_7DkU · 2024-08-10
> > **thank you**
> >
> > I'd like to thank the authors for their rebuttal which I found useful to address my concerns.
> > I feel that the revision required to account for including the ablations, moving parts of the appendix to the main paper and address all the other comments will be substantial. Because of this I've only slightly increased my score.

---

> > > ### Author Response · Authors · 2024-08-13
> > > **Thank you**
> > >
> > > We thank the reviewer for their time and reply. We are glad that we are able to improve their impression of this work.

---

### Official Review · Reviewer_8dkQ · 2024-07-14

**Soundness:** 3
**Presentation:** 2
**Contribution:** 3
**Rating:** 6
**Confidence:** 3

**Summary:**

In this work, the authors proposed CordsNet (Convolutional RNN dynamical system), which provides incorporates a conventional neuroscientific model of RNN incorporating the convolutional layer.

**Strengths:**

1. The authors analyzed the proposed dynamic convolutional RNN from different aspects.

2. Derived batch normalization for the case of linear dynamic RNN

**Weaknesses:**

1. The authors failed to compare it with a series of works named Deep Equilibrium Models (DEQs), which model a layer or a block of layers including the non-linearities as a fixed-point problem.  CordsNet is limited to the linear part of the layer while DEQs are not. And DEQs achieved SoTA performance on many large-scale tasks.

2. In the comparison seems Convolutional LSTM/GRU is missing, which is an important family of RNN with Conv layers. Could the authors explain why they are not necessary to be included in the benchmarks?

3. The contribution points are really scattered and don’t feel elaborated enough. Maybe it’s more suitable as a journal paper with more elaborated experimental details for each section.

**Questions:**

1. Is the analytical formulation of CordNets limited to only linear layer?

**Limitations:**

The authors discussed the limitations.

---

> ### Comment · Reviewer_7SU8 · 2024-08-03
>
> This reviewer (8dkQ) seems to not understand the field of "modelling the primate visual cortex". I suggest you to read this paper: https://www.nature.com/articles/s41593-019-0520-2 before making a new judgement. The proposed method is definitely novel because there is no dynamical model of CNN that can model visual cortex before. The proposed method definitely contributes to computational neuroscience. The proposed method also shows some interesting properties, e.g., in Fig. 5.

---

> ### Author Rebuttal · Authors · 2024-08-07
>
> We thank the reviewer for raising these important points in their review. Our general interpretation of the reviewer's concern is that this work failed to provide what seems to be minimally-required benchmarks, which in turn translates to lack of evidence for any performance improvements. The reviewer is also well-read in the field of convolutional, recurrent and fixed-point architectures, and thus was able to pinpoint key literature that overlap with our proposed work. From this point of view, it can be hard to find significance in this work.
>
> As such, we would like to provide a personalized summary of this work that brings the reviewer's aptly-raised concerns into perspective and explains why we believe our work is still a significant contribution despite those concerns.
>
> - **Dynamical systems in neuroscience** study how neural activity evolves over time and how these changes relate to brain functions. They are prevalent in neuroscience, recently in the form of continuous-time RNN dynamical systems with biological constraints such as realistic neuron time constants.
> - **Training with a handicap.** Such dynamical RNNs are harder to train than vanilla RNNs in machine learning, because of the continuous-time property that inflates the number of time steps when coupled with the fast time constants present in biological neurons. Since vanilla RNNs are already ill-suited for image processing, this makes dynamical RNNs even harder to train on ImageNet, and for this reason no such model has been proposed to date.
> - **Motivation and potential.** Yet, there is a strong need for such a model to be built, so that decades of dynamical systems theory can be applied to a model that can actually process natural images. Such a model would be studied by the theoretical neuroscience community to understand continuous-time dynamics underlying biological visual information processing, which we hope will give rise to new ideas and theories to be tested.
> - **Defining expectations.** To that end, our objective is to train the first dynamical system to classify ImageNet at an accuracy that would be deemed as “comparable to CNNs”. We approach this goal with the understanding that the constraints we impose on our model are likely to be detrimental to performance, and we will most likely not be achieving anything SoTA outside the field of neuroscience at this stage; we therefore omit benchmarks typically found in machine learning literature.
> - **Significance of our architecture.** CNNs and convolutional RNNs have already been proposed and extensively analyzed in vision neuroscience as candidate models of the visual system. The logical approach would therefore be to build a continuous-time dynamical extension of these models. Such a model would not only build on the work on convolutional architectures by the vision neuroscience community, but also incorporate the legacy of dynamical systems from the wider neuroscience community.
> - **Evaluating our success.** The best top-1 accuracy achieved by our models on ImageNet (<60%) is modest compared to what vision models today can do. Understandably, as a result of the aforementioned biological constraints, our models fail to outperform basic CNNs with the same convolutional layers (but we are close). But ultimately, we can reasonably say that the models that we trained are indeed performing some meaningful processing of visual information, and that we have successfully built a dynamical system that can classify natural images.
> - **Focusing on the objective.** Finally, we dedicated the majority of the main text, including figures 3,4,5 and 6, to show how our trained model can be studied as a continuous-time dynamical system in various fields of neuroscience, thereby highlighting its potential applications and how it opens up new avenues of research in these areas.
>
> We hope that this summary is able to help the author gain a different (and hopefully more optimistic) perspective of our work. We now address specific points raised by the reviewer below.
>
> **Issue on DEQs**
>
> We thank the reviewer for bringing DEQs into the discussion. Due to the character limit, we refer the reviewer to our discussion on DEQs in the general response.
>
> **Issue on linearity**
>
> CordsNet is nonlinear in the exact same way that RNNs or DEQs are nonlinear. The evolution of activity across time in CordsNet is described by equation (1) in the main text, which includes a ReLU activation function. While training the nonlinear model, we split training into several stages (Figure 2B), and one of the stages involves linear models (lines 142-152), which may be the source of this misunderstanding.
>
> **Comparison with convolutional LSTM/GRU**
>
> There are two key reasons why comparison with gated RNNs is not necessary.
> - Artificial gating mechanisms are not biologically-realistic and do not fall within our objectives.
> - We are trying to model the biological visual system, which is (generally) not the part of the brain that stores memories or makes decisions based on long-term dependencies. Our task is classifying ImageNet, where gating mechanisms may not offer much advantage.
>
> **Issue on scattered points**
>
> We thank the reviewer for the suggestion. Overall, we feel that this work lies comfortably in the intersection of machine learning and neuroscience, and that there would be an audience for this work in this conference. We also acknowledge the reviewer’s concern that the work presents too many points that have not been sufficiently covered. CordsNet is both a dynamical RNN and a CNN, where each architecture, on their own, has been studied by the neuroscience community for years. As such, the narrative of our paper is focused on introducing the model, how it compares with past RNN or CNN models in neuroscience, and briefly demonstrating the many ways in which new research directions can be explored using this hybrid model. It is also why we developed tools to help analyze our model (lines 214-216).

---

> > ### Comment · Reviewer_8dkQ · 2024-08-11
> >
> > I thank the authors for the detailed rebuttal which convinced me comparison with sota computer vision models is not necessary in this work. The work is more related to the CORNet works and in that context I can better appreciate the contributions. Thus I am raising the ratings.
> >
> > I am also curious about whether such modelled architectures have advantages over artificial neural networks such as in (adversarial) robustness. But this is out side the scope of this work I understand.

---

> > > ### Author Response · Authors · 2024-08-13
> > > **Thank you**
> > >
> > > We thank the reviewer for taking the time to better understand our work and appreciating our results from a different angle.
> > >
> > > *(For meta-reviewing purposes, we would like to state that everything below is not related to the rebuttal and is simply an out-of-scope discussion with the reviewer.)*
> > >
> > > The reviewer made a sharp observation by considering the advantages of biological visual perception (which is essentially immune to the kind of adversarial images that plague artificial models), there might be some potential to exploit this and improve artificial models. In general, biological perception differs from machine perception in the following ways:
> > > - context awareness, where the entire image is interpreted as a whole and the subject can predict what might happen next or relate what they are seeing with past experience
> > > - biological attention mechanisms, in which subjects can focus on certain stimuli by adjusting their brightness sensitivity and maintaining fixation
> > > - 3D perception, where a 2D input image is understood as a 3D scene in the mind of a subject
> > > - multisensory integration, where hearing and smell can influence a visual percept
> > >
> > > among many others. These factors contribute to biological holistic scene understanding, leading to robustness. The reviewer is right that there is potential for biologically-motivated models that incorporate any of these effects to be a contribution for the machine learning community. As a pure speculation, we think early implementations of these effects would make training harder, and may result in a robust model but with lower classification performance, just like in our work. The common theme of all the aforementioned mechanisms is that they all need time to process in a biological brain, which is why we hope that our work can inspire these types of research directions in the future.

---

### Official Review · Reviewer_A7p2 · 2024-07-15

**Soundness:** 4
**Presentation:** 3
**Contribution:** 4
**Rating:** 8
**Confidence:** 3

**Summary:**

Inspired by neuroscience, the authors propose to introduce recurrent connections in conventional convolutional neural networks (CNNs). The resulting model shows comparable performance with regular CNNs, but exhibits higher noise robustness. The authors also developed a toolkit to analyze the resulting architecture, using iterative and dimensionality reduction methods. The architecture is used to model complex cognitive tasks, mimicking the higher order visual areas in the monkey.

**Strengths:**

- This paper seamlessly weaves inspiration from neuroscience, dynamical systems theory, learning algorithms, analysis methods, performance, and modeling and replicating results from neuroscience experiments.

- Introducing continuous dynamics into CNN+RNN seems to exhibit really interesting properties.

- Analysis of the dynamical behavior by finding the eigenvalues directly from the recurrent weight matrices is interesting. This has been used to uncover some really interesting dynamical patterns dependent on the time of classification (fater or slower inference).

- The use of the model in various visual cognitive tasks (in V4 and IT) is also quite impressive.

**Weaknesses:**

- Too much technical details are hidden in the appendix. Please at least provide a brief sketch in the main text.

**Questions:**

Questions

- Isn't the task in Fig 5D handled by area MT? Have you considered computing the BrainScore for this? In general, can results in Fig 5 be compared to the experimental literature?

Comments

- There is an earlier recurrent CNN than the one cited: Ming Liang and Xiaolin Hu. Recurrent convolutional neural network for object recognition. In Proceedings of the IEEE conference on computer vision and pattern recognition, pages 3367–3375, 2015.

**Limitations:**

The discussion/conclusion provides adequate assessment.

---

> ### Author Rebuttal · Authors · 2024-08-07
>
> We truly appreciate the reviewer for the encouraging and positive review.
>
> **Technical details in the appendix**
>
> The reviewer has expressed concern that important technical details that are in the appendix should briefly be mentioned in the main text. We agree with this point and refer the reviewer to the general rebuttal for all the changes we plan to make to our submission.
>
> **Random dot motion task and area MT**
>
> The reviewer has correctly mentioned experimental literature pertaining to evidence of neural activity tuned to random dot direction and coherence in area MT in the dorsal stream. In general, the RNN that we append to the back of our model represents either a decision-making part of the brain, such as the prefrontal cortex, or a motor region such as the frontal eye fields (which is still prefrontal). While MT contains direction-tuned cells, the eventual motor response (in the form of eye saccades) would likely be similarly tuned. However, this is ultimately an (unsubstantiated) interpretation on our end. A more rigorous analysis is required for any conclusions to be drawn.
>
> **Interpretation of results in Figure 5**
>
> The main goal of the results in Figure 5 is to illustrate the point that we can now build completely continuous-time models that accept real images as an input, rather than abstract one-hot vectors as found in many past works. We do provide some interpretation of the solutions we found, but to draw any conclusions about the brain from this would require a much more in-depth analysis that is currently outside the scope of this work. We do not have the neural data to compute BrainScore, and we foresee that the dataset would not be big enough (small number of coherence levels) for strong statistical significance.
>
> **Missing citation**
>
> We thank the reviewer for pointing out this missing citation, we will add this citation into our introduction. We will also thoroughly find and include other relevant literature in our final version.

---

> > ### Comment · Reviewer_A7p2 · 2024-08-09
> > **Detailed comments appreciated**
> >
> > Thank you for the detailed response. The proposed revision plan also looks reasonable.

---

> > > ### Author Response · Authors · 2024-08-13
> > > **Thank you**
> > >
> > > We thank the reviewer for the acknowledgement and once again would like to express our appreciation for the positive review.

---

### Author Rebuttal · Authors · 2024-08-07

We thank all reviewers for their time and effort in reviewing our submission. We will address common issues here.

**Important details in Appendix**

Reviewers A7p2 and 7DkU raised the issue that important results are reported in the appendix, and recommended for their (minimally) brief inclusion in the main text. We agree with this point, and provide a summary of all the changes to our submission below. This summary includes additional changes requested from other reviewers as well.

1. Introduction (additional 3-4 lines)
- mention existing works on recurrent CNNs in machine learning literature
- briefly mention DEQs and their relevance to this work (complete review in the appendix)

2. Model architecture (additional 15-20 lines)
- introduce the model architecture in a more complete way
- include details of the analysis from Appendix B

3. Training and results (additional 5-10 lines)
- move the technical derivations to the appendix (lines 139-156) and briefly describe them in the main text
- include brief descriptions and state results from the ablation studies in the appendix
- move parts of Table S5 into the main text and interpret the results

4. Model analysis (reduced by 15-20 lines)
- remove Figure 3A as Figure 3B illustrates a similar point
- remove explanations about Figure 3A

5. Applications  (reduced by 5-10 lines)
-  remove the methodology for BrainScore in the main text (lines 236-242) and instead elaborate in detail as a section in the appendix

These changes are expected to stay within the 9-page limit. However, if our work is accepted, we would also make full use of the additional page to remedy the information load in the appendix.

**Comparison to DEQs**

Reviewer 8dkQ commented that comparing our model to DEQs is necessary and important. DEQs are inherently similar to RNNs, and therefore CordsNets by transitivity. To address this point, we will mention DEQs in the introduction in the main text, as well as a more in-depth review of the differences between CordsNets and DEQs as a subsection in Appendix A. Fundamentally, DEQs are focused on arriving at a fixed point required for the completion of some task. In contrast, we proposed CordsNet as a model of the biological visual system, and we are concerned with the model activations before, during, and after steady-state inference has been achieved, so that we may compare them with experimental data and generate new hypotheses on how the brain works.

- CordsNet expresses a range of dynamical behaviors depending on the required task, including oscillatory, chaotic and stable dynamical regimes (Figure 1B). Even within the stable regime, which results in fixed points, they manifest in different patterns, such as point, line and ring attractors (Figures 1C, 1D and S3). Evidence of these temporal behaviors have been found in the brain, which is the focus of this work, even though they may not be ideal for tasks. On the other hand, DEQs prefer fixed points only, in the true spirit of machine learning that seeks optimal performance.

- When DEQs complete a task, the simulation ends. They are run again from scratch for the next task. In contrast, a biological brain runs perpetually, even when there are no tasks. CordsNet follows the same principles, and returns to some baseline activity by itself without external interference after the input image is removed (Figure 3B). Another image can then be presented for future inference.

**Ablation studies**

Reviewers 7DkU and UEUJ have requested for certain ablation studies to be performed in order to justify several design choices in our models. We carefully disect each term in our loss function, as shown in equation (1) of the rebuttal PDF.  Firstly, we note that the goal of our model is to bring together the results from CNN vision neuroscience and RNN cognitive neuroscience communities. To that end, we have to select a suitable objective that is guided by experimental literature (Figure R1 in rebuttal PDF). This influences the inference window in which we aim to minimize the cross-entropy classification loss, and also determines the neuron time constant of 10ms, as well as the way stimuli is presented to the model.

We next look at the effects of introducing the spontaneous penalty term (Figure R2). In order to do so, we train 20 CordsNet-R2s on the CIFAR-10 dataset for 6 different values of penalty coefficients, for a total of 120 models in this analysis. Without the spontaneous penalty term, we find that the solutions can fall in three broad dynamical regimes: unstable, multistable and monostable (Figure R2A). In the unstable regime (top row), neural activity explodes after stimulus presentation, and never recovers, which is undesirable. In the multistable regime (middle row), the model remains stable throughout the first stimulus presentation, but is also unable to return to baseline due to the presence of additional attractors in its activity space. As such, when a second image is presented, it is unable to make the correct classification, despite being stable. The only solution that we want is the monostable case. Figure R2B presents the effects of different penalty coefficients on the types of solutions obtained. We find that our choice of $10^{-3}$ is ideal for only allowing monostable solutions.

We also performed an ablation study on the log-weights that we have introduced to our cross-entropy term. Here, we trained 10 CordsNet-R2s for 5 different log-weighing scales. We find that without this term, there is a small chance that a transient solution is obtained (Figure R3B), where the model does not arrive at a fixed point, but is instead optimized to only classify for the particular inference window. As such, it is unable to exhibit most of the other properties that our models in the main text have. By introducing this weighing scheme, we have eliminated this class of solutions.

---

### Public Comment · ~Sushrut_Thorat1 · 2024-11-23
**Additional links to previous literature**

While it is always nice to see much-needed new work in the domain of RCNNs, it is disappointing to see critical parts of literature not represented in positioning this work, potentially overselling the impact here. I'd like to point the authors to such literature in this comment.

"we propose a hybrid architecture that integrates the continuous-time recurrent dynamics of RNNs with the spatial processing capabilities
of CNNs, which we refer to as CordsNet (Convolutional RNN dynamical system)" - this marriage between RNNs and CNNs has been around since a long time - as another reviewer pointed out. You do cite and compare against CORNet-RT, however there is a, arguably more natural, class of RCNNs that Tim Kietzmann's group has been working on for quite some time, e.g. see: https://www.frontiersin.org/journals/psychology/articles/10.3389/fpsyg.2017.01551/full ; https://journals.plos.org/ploscompbiol/article?id=10.1371/journal.pcbi.1008215

"Training a continuous-time model is computationally expensive. We propose a computationally cheaper algorithm" - you might want to check out Drew Linsley's work on training RNNs efficiently - https://proceedings.neurips.cc/paper/2020/hash/766d856ef1a6b02f93d894415e6bfa0e-Abstract.html ; additional note: RCNNs can be trained at scale (ecoset/MS-Coco, e.g. https://arxiv.org/abs/2209.11737v2; for architecture definition see e.g. https://github.com/KietzmannLab/BLT-pytorch-CCN23) without any hiccups. I see that in your case you need to train for 100s of timesteps and typical training regimes make little sense - however I wonder how much of that "timestep expressivity" is actually necessary. Usually what we find is 4-5 timesteps of a deep RCNN are sufficient to fully process imagenet/ecoset images - if you modified your algorithm to only unroll the RCNN for a limited # of timesteps and then artificially interpolate between them, would your results be any worse, I wonder. Additionally, the best imagenet top-1 acc you report is 41.2% in Fig. 2. - it is straightforward to get around 70% top-1 with the RCNN architecture mentioned above (wherein the corresponding feedforward CNN can barely hit 60%) - which brings the expressivity of the architecture being presented in this paper into question.

"Analyzing our architecture as a dynamical system is computationally expensive, so we develop a toolkit consisting of iterative methods
specifically tailored for convolutional structures." - on analyzing RCNNs you might find these papers relevant - https://openreview.net/forum?id=BJpv46DGNl_ ; https://arxiv.org/abs/2308.12435 (for a review of general trends, see: https://www.sciencedirect.com/science/article/pii/S0959438820301768) - there have been steps taken in the field in analyzing the internal dynamics of RCNNs.

"In monkey neural recordings, our models capture time-dependent variations in neural activity in higher-order visual areas." - please check out Tim Kietzmann's seminal work on this front: https://www.pnas.org/doi/abs/10.1073/pnas.1905544116 , where they used a scaled-up RCNN to predict MEG activity across brain regions and time! Highly relevant.

"We also train multi-area RNNs using our architecture as the front-end to perform complex cognitive tasks previously
impossible to learn or achievable only with oversimplified stimulus representations." - check out recent work by Saeed Salehi: https://www.biorxiv.org/content/10.1101/2024.09.09.612033v2 where they also train a scaled-up RCNN (with a 2-stream model inspired by cortical computations) to function for a multitude of visual "cognitive" tasks.

It is unfortunate that the reviewers didn't catch many of these crucial references, however let this comment be a reminder that they exist, thereby making the contribution of this work clearer by both supporting its claims and highlighting what's new.

---

> ### Public Comment · ~Wayne_WM_Soo1 · 2024-11-23
> **Addressing comments**
>
> Hi and thank you for your interest in this work! It is important that our work reaches the neuroscience RCNN community, because we are presenting a new architecture that aims to be alternatives to RCNN for the same exact uses. We will address the references in a separate comment, but we believe that a single point will largely address the key misunderstanding here.
>
> We would like to point out that none of the references cited are related to continuous-time RNN dynamical systems (which is basically the title of our paper). Some early references of such an architecture are from around 30 years ago (these references are all in the paper, numbered accordingly):
>
> [3] 10.1126/science.274.5293.1724
>
> [5] doi.org/10.1073/pnas.93.23.13339
>
> [7] doi.org/10.1523/JNEUROSCI.16-06-02112.1996
>
> A continuous-time RNN dynamical system is typically characterized by:
> 1. a leaky term (the -r in equation 2)
> 2. biological neuron time constants (the T in equation 2)
> 3. continuous-time smooth trajectories (the dr/dt in equation 2)
> 4. no normalization (although this can easily be explored)
>
> These properties set dynamical RNNs apart from machine learning RNNs. They are also harder to train because having smooth, continuous trajectories inflate the number of time steps, but they also have cool dynamics such as oscillations, chaos, fixed points, mono- and multi-stability (see Figure 1B-C, S1 and S4). These are the dynamics that we want to bake into CNNs. In fact, one of the references that you cited actually made this distinction, citing Dayan and Abbott:
>
> doi.org/10.1016/j.conb.2020.11.009 in a section titled "Continuous-time versus discrete-time dynamics"
>
> In the paper, we referred to what you call RCNNs as discrete-time CNNs (DT-CNNs), because they lack the properties mentioned above, and dedicated almost an entire figure to differentiate ourselves from this architecture in Figure 3. So training for optimal ImageNet performance over 4-5 timesteps is never the point of this work.
>
> **Speaking for myself, I hope that you can review Figures 1, 3 and S1,2,3,4 thoroughly and gain a different perspective about our work**, and it would particularly address the following question:
>
> "if you modified your algorithm to only unroll the RCNN for a limited # of timesteps and then artificially interpolate between them, would your results be any worse, I wonder"
>
> Finally, the reviewers are aware of the existence of CORNet, arguably one of the most famous works of RCNN in neuroscience, and have agreed that this work is a contribution beyond that work.

---

> ### Public Comment · ~Wayne_WM_Soo1 · 2024-11-23
> **References**
>
> In this comment, we will address the references.
>
> Existing RCNN papers
>
> These are important references that we will add to our paper as soon as the editing window is up. They will largely go into the introductory text (specifically the top of page 2), where we already have made reference to a few other recurrent CNN papers. These include:
>
> https://www.frontiersin.org/journals/psychology/articles/10.3389/fpsyg.2017.01551/full
> https://journals.plos.org/ploscompbiol/article?id=10.1371/journal.pcbi.1008215
> https://www.pnas.org/doi/abs/10.1073/pnas.1905544116
> https://www.sciencedirect.com/science/article/pii/S0959438820301768
> https://arxiv.org/abs/2209.11737v2
>
> [1] https://proceedings.neurips.cc/paper/2020/hash/766d856ef1a6b02f93d894415e6bfa0e-Abstract.html
> This work is great and may have potential uses for continuous-time models. We will add that to our conclusion.
>
> [2] https://arxiv.org/abs/2308.12435
> As we have claimed, "Analyzing our architecture as a **dynamical system** is computationally expensive, so we develop a toolkit consisting of iterative methods **specifically tailored for convolutional structures**." Analysis of an RNN dynamical system typically involves studying the recurrent weight matrix, which in this case is a convolution. Speaking for myself, I do not see any indication of dynamical systems or tailoring towards convolutional structures in the proposed reference. Nonetheless we will also cite this work as it is relevant.
>
> [3] https://www.biorxiv.org/content/10.1101/2024.09.09.612033v2
> Regarding applications in general, we are positioning our work as potential alternatives to CNNs (in the context of Figure 5) and RCNNs (in the context of Figure 6). There are many instances of CNNs being applied in neuroscience, as we mentioned in the second sentence of our abstract, and it would be hard to cite them all. Also, this particular reference seems to be quite recent in late 2024 (in light of the fact that the submission deadline for this conference is in May 2024), but we will also cite it as it is clearly relevant to our work.

---

> ### Public Comment · ~Sushrut_Thorat1 · 2024-11-23
> **Clarification**
>
> Thank you for reaching back. Although I did/do understand that extending RCNNs to the continuous time domain (what many would call "actual" recurrence) is novel, discrete RCNNs are a related class which is well-studied and I don't understand how the insights found in that literature wouldn't speak to the continuous time domain.
>
> "The activity in CORNet-RT varies over time, and spikes briefly at the time step in which it is trained to make an accurate classification of the input image. It is not able to make an accurate prediction at all other times." - Training a RCNN to output the class only at the last timestep is one way to train them. In the other class of models (Spoerer et al. and also the other papers I mentioned) they are trained to output the correct class over all timesteps. This provides interesting abilities to the RCNN wherein even if trained only for N timesteps, it can maintain its classification accuracy ad-infinitum as long as the stimulus is presented (see Fig.7 in https://arxiv.org/abs/2308.12435). It would be interesting to check how these models behave in the settings in Fig. 3.  I reckon the profiles would look quite different from CORNet-RT, although they might still differ from your network - but that would be interesting.
>
> Furthermore, there's no reason that dynamics such as oscillations, chaos, fixed points, mono- and multi-stability cannot arise in regular RCNNs given the right objectives. Fixed points / limit cycles are possibly easy - for e.g. Fig. 2 of https://arxiv.org/abs/2308.12435 already shows the change in representation reducing over time. If the other phenomena are essential for the computations required in a task I apriori see no reason why they can't just emerge in RCNNs (non-linear dynamics in discrete spaces do produce all these phenomena).
>
> "So training for optimal ImageNet performance over 4-5 timesteps is never the point of this work." This is fair - I was not trying to say that you need to beat the state-of-art but was trying to hint at scalability - I should've been more precise. This relates to my Q: "if you modified your algorithm to only unroll the RCNN for a limited # of timesteps and then artificially interpolate between them, would your results be any worse, I wonder" - this would be similar to having larger "steps" in solving your differential equation - but I understand that you want to emphasize the dynamics and not worry too much about accuracy for now. This point was triggered by the abstract wherein you stated "Our models preserve the dynamical characteristics typical of RNNs while having comparable performance with their conventional CNN counterparts on benchmarks like ImageNet." which one can also read as - "this will scale similarly to the counterpart CNNs on Imagenet".
>
> In ref [2], the network is a RCNN - it is convolutional and recurrent and what we analyzed are dynamics. Dynamics can exist in systems which are not mathematically designed to follow the usual state transition equations like Eq. 2 in your paper., and analysis of the dynamics can also take on various forms. This is related to what I'm generally trying to communicate with my comments - that there have been interesting, complimentary, developments in the DT-RCNN space that seem to be closely aligned with many of the interests of this paper.
>
> Ref [3] is very new indeed - it was meant as a reading suggestion - actually all the references were. I thought the camera-ready version was already out which is why I posted a comment instead of writing an email to you - so that these references stay attached here as a comment.
>
> I apologize if the initial comment came across as too aggressive - I just feel that research in both continuous and discrete time RCNN share the same goals and that they should proceed in tandem. Please let me know if I am still mischaracterizing something.

---

> ### Public Comment · ~Wayne_WM_Soo1 · 2024-11-23
> **Thank you**
>
> Those are great points being made -- we will briefly reply some specific points and end with a final general statement.
>
> - Continuous vs discrete-time: There are many cases where discrete-time models behave just like continuous-time models, and just as many that do not due to discretization errors. For example, simulating dx/dt = sin(x) for too large of a $\Delta t$ would give you a wrong solution.
>
> - RCNNs in Figure 3: Discrete-time models can potentially achieve the same effects in the figure (I acknowledge Figure 7 and I am curious as well). However, this has been known theoretically for decades in dynamical RNNs, see Chapter 7.4 of Dayan and Abbott. This is an example of a long-known result from dynamical RNNs being applied to RCNNs, which is the point of this work.
>
> - ImageNet discussion: Relevant results are in Table 1, in the last two columns, where we compare fine-tuned CordsNets to their static CNN counterparts without recurrence, showing comparable performance. This is the result the abstract is referencing (it scales from R2 to R8 at least).
>
> Overall, we are proposing a hybrid model between continuous-time RNNs and (R)CNNs. Our audience would be split between the two subfields. The RCNN community can ask questions like "*isn't discrete-time good enough?*", just like how the dynamical RNN community will ask "*is the brain actually performing convolutions?*" which is arguably more controversial. If this work can generate such discussions, then we believe we have contributed something to the field.

---

> ### Public Comment · ~Sushrut_Thorat1 · 2024-11-23
> **Final response**
>
> Thanks for engaging in this discussion - I’ll edit the title of the first comment so it doesn’t come across as negative to someone checking out the paper.
>
> “If this work can generate such discussions, then we believe we have contributed something to the field.” I agree. Looking forward to new ideas on this continuous-time - discrete RCNN continuum. I believe the aforementioned references and this discussion will anyway benefit anyone reading your paper.

---

### Decision · Program_Chairs · 2024-09-25

**Decision:**

Accept (spotlight)

**Comment:**

There are abundant recurrent connections in biological circuits. Although there have been multiple neural network models that incorporate such recurrent connectivity, non-recurrent networks still constitute the mainstream of applications in computer science and there is ample room for improvement.
Reviewers were mostly enthusiastic about the current work.
Many of the concerns could be addressed during the revision before submitting the final version, including:
Citation and discussion of other work with recurrent neural networks, and potentially comparison with other benchmarks (see especially specific suggestions from 8dkQ, UEUj, 7SU8, and 7DkU).
Reviewer 7DkU rightly noted that it would be important to provide more technical details, but I think that it would be easy for the authors to provide this information before the final version.
The authors provided clear and rigorous responses to the questions and should be able to improve the presentation in the final version.
I share the reviewers’ enthusiasm for this paper.